# Brain Network Abnormalities in Obsessive–Compulsive Disorder: Insights from Edge Functional Connectivity Analysis

**DOI:** 10.3390/bs15040488

**Published:** 2025-04-08

**Authors:** Yongwang Xu, Hongfei Liu, Haiyan Liu, Defeng Lin, Sipeng Wu, Ziwen Peng

**Affiliations:** 1Center for the Study of Applied Psychology, Guangdong Key Laboratory of Mental Health and Cognitive Science, School of Psychology, South China Normal University, Guangzhou 510631, China; xyongw@scnu.edu.cn (Y.X.); 2023023866@m.scnu.edu.cn (H.L.); 2Key Laboratory of Brain, Cognition and Education Sciences, Philosophy and Social Science Laboratory of Reading and Development in Children and Adolescents (South China Normal University), Ministry of Education, Guangzhou 510631, China; 3School of Artificial Intelligence, South China Normal University, Foshan 510631, China; 2023025174@m.scnu.edu.cn (H.L.); lindefeng@m.scnu.edu.cn (D.L.); 4Aberdeen Institute of Data Science and Artificial Intelligence, South China Normal University, Guangzhou 510631, China; u18sw21@abdn.ac.uk

**Keywords:** obsessive–compulsive disorder, edge functional connectivity, network entropy

## Abstract

Functional differences in key brain networks, including the dorsal attention network (DAN), control network (CN), and default mode network (DMN), have been identified in individuals with obsessive–compulsive disorder (OCD). However, the precise nature of these differences remains unclear. In this study, we further explored these differences and validated previous findings using a novel edge functional connectivity (eFC) approach, which enables a more refined analysis of brain network interaction. By employing this advanced method, we sought to gain deeper insights into FC alterations that may underlie the pathology of OCD. We collected data during movie watching from 44 patients with OCD and 33 healthy controls (HCs). The two-sample *t* test was used to assess differences in entropy between the DAN, CN, and DMN between groups. The analysis was performed with control for potentially confounding variables to ensure the robustness of the findings. Significant differences in network entropy were found between the OCD and HC groups. Relative to HCs, patients with OCD showed significantly reduced entropy in the DAN and increased entropy in the CN and DMN. The decreased entropy in the DAN and increased entropy in the CN and DMN observed in this study may be related to the core symptoms of OCD, such as attention deficit, impaired cognitive control, and self-referential thinking. These results provide valuable insights into the neurobiological mechanisms of OCD and highlight the potential of network entropy as a biomarker for the disorder. Future research should further explore the relationship between these network changes and the severity of OCD symptoms, as well as assess their implications for the development of treatment strategies.

## 1. Introduction

Obsessive–compulsive disorder (OCD) is a chronic neuropsychiatric disorder characterized by persistent intrusive thoughts (obsessions) and repetitive behaviors (compulsions) aimed at alleviating anxiety, which have profound impacts on the lives of patients and their families ([3]; [61]; [28]). Recent studies suggest that OCD may involve functional abnormalities in large-scale brain networks, particularly those associated with cognitive control, response inhibition, and executive functions ([1]; [14]; [23]). OCD patients often exhibit deficits in motor initiation, execution, and inhibition, as well as impairments in spatial working memory and spatial recognition abilities ([43]; [42]; [45]). Notably, these network dysfunctions are not only observed in OCD patients but also in their first-degree relatives, suggesting the presence of potential endophenotypic markers ([9]; [8]; [56]). OCD affects approximately 2–3% of the global population, highlighting its prevalence and the urgent need for a deeper understanding of the neural mechanisms underlying the disorder. Given the observed functional and structural abnormalities in multiple brain networks, it is also critical to develop effective intervention strategies to better support those affected by OCD ([30]).

Currently, the diagnosis of OCD relies primarily on the Diagnostic and Statistical Manual of Mental Disorders and International Classification of Diseases criteria, and it is often influenced by subjective judgment due to the lack of objective biomarkers ([10]). This diagnostic approach can lead to misdiagnosis or underdiagnosis, especially given the high rate of comorbidity with other conditions, such as depression and anxiety. Furthermore, the complexity and heterogeneity of OCD add to the challenges of accurate diagnosis ([19]).

In recent years, node-based functional connectivity (FC) has been used to model brain networks, examining the functional coupling between different brain regions to investigate the neural mechanisms underlying OCD. They also show notable abnormalities in the connectivity between the prefrontal cortex and basal ganglia and functional activity in the cingulate cortex and other brain networks, such as the default mode network (DMN) and executive control network. These alterations are related closely to OCD symptoms, including impaired self-regulation, impulse control, and emotional processing ([21]). Changes in the functional connectivity of the fronto-striatal and fronto-parietal networks are believed to be closely related to cognitive rigidity and compulsive behaviors in OCD patients, which are among the core features of the disorder ([40]; [55]; [15]). Patients with OCD also exhibit hyperconnectivity between the dorsal attention and frontoparietal networks, which is associated with the treatment response; greater connectivity correlates with greater symptom improvement following treatment, reflecting effects on neural mechanisms underlying attention control, cognitive flexibility, and executive function ([2]). Moreover, they have abnormal FC between the DMN (associated with rest and self-reflection) and occipital cortex (involved in visual processing) that may contribute to OCD symptoms, such as excessive self-focus and intrusive thoughts, which in turn may affect the execution of visual and cognitive tasks and exacerbate symptoms ([6]; [5]; [22]). In addition, network analysis techniques have been used to explore cognitive dysfunction in both healthy individuals and clinical populations. Studies on response inhibition, proactive, and reactive control mechanisms in OCD and related disorders (such as ADHD) suggest that network dysfunctions may lead to executive function deficits, compulsive behaviors, and motor impairments ([4]; [65]; [39]; [29]). Moreover, neuroimaging studies have shown that dynamic functional connectivity changes in OCD patients are closely related to the severity of their symptoms and cognitive performance ([24]; [55]; [41]).

An early study by [31] ([31]) found that patients with OCD had abnormal functional connectivity in the default mode network (DMN), and its hyperactivity was closely related to intrusive thoughts. This conclusion has been further supported in subsequent studies. For example, [26] ([26]) found through task-based fMRI that the hyperconnectivity of the DMN was significantly associated with patients’ compulsive introspection, while [58] ([58]) used dynamic functional connectivity analysis to reveal the dynamic association between the dysfunction of the cortical-striatal circuit and symptom severity. These studies together suggest that the dysfunction of the DMN may be one of the core features of the pathological mechanism of OCD.

In the fields of behavioral tasks and cognitive abilities, research has clearly shown that patients with OCD have significant deficits in response inhibition and attention control. [63] ([63]) found through the Go/No-Go task that the abnormal activation of the prefrontal–parietal network in patients was directly related to the decline in inhibitory control ability. [35] ([35]) further revealed the predictive role of the functional connectivity strength of the basal ganglia–prefrontal circuit in response inhibition efficiency during the stop-signal task. In the stop-signal task, participants are required to immediately inhibit their current action or response upon receiving a stop signal, which demands the individual’s ability to flexibly control behavior in response to unexpected interrupt signals. This task is widely used to assess an individual’s response inhibition ability, particularly evaluating their efficiency in quickly suppressing impulsive behaviors. The classic study by [12] ([12]) emphasized the central role of the prefrontal–parietal network in top–down attention regulation, but its specific mechanism of action in OCD has not been fully elucidated. It is worth noting that most existing studies analyze a single cognitive dimension (such as inhibitory control or attention allocation) in isolation, lacking an integrated analysis of the multi-task synergy effect and dynamic network reorganization, which limits the integrity of the cognitive neural model.

As research gradually shifts towards the exploration of multi-network interactions and dynamic characteristics. [50] ([50]) proposed that dysfunction of the dorsal attention network (DAN) may lead to difficulties in attention shifting, and a systematic review by [17] ([17]) pointed out that abnormal functions of the control network (CN) may exacerbate behavioral rigidity by weakening executive functions. [60] ([60]), through their study of the information-processing dynamics in neural networks, further revealed the association between the default mode network (DMN) and compulsive symptoms. This association may stem from an imbalance in the competitive interactions between the DMN, the dorsal attention network (DAN), and the control network (CN). In this context, emerging entropy analysis provides a new tool for quantifying the dynamic complexity of networks. Metrics such as transfer entropy and network entropy have been used to assess the brain’s adaptability in cognitive tasks, providing new insights into impaired executive control mechanisms in OCD and related disorders ([66]). Research using the widely applied stop-signal task to assess inhibitory control has found significant changes in both functional connectivity and entropy dynamics in OCD patients ([49]; [52]). For example, [48] ([48]) found that abnormal changes in the entropy of the brain network in OCD patients were related to a decrease in information integration efficiency, and a study by [54] ([54]) suggested that the functional connection strength of the cortical–limbic system could predict cognitive flexibility impairment.

Node-based connectivity studies have revealed functional differences across multiple brain networks in patients with OCD, providing further support for the identification of the biological markers of the disease. However, node-based connectivity approaches have certain limitations. In recent years, edge functional connectivity (eFC) analysis has been proposed to overcome the shortcomings of traditional node-based methods. This approach provides more detailed information about FC than conventional biomarker-based approaches do and is particularly useful for the exploration of interregional interaction and multidimensional network dynamics ([32]; [11]; [59]). It effectively enables the understanding of the complexity of brain functional networks and provides a more refined and dynamic perspective, especially on the network bases of neuropsychiatric disorders.

The brain is an extremely complex and highly interconnected network system, in which the connections between nodes (edges) play a crucial role in information transmission and integration. Network entropy, as a key indicator for measuring the complexity of information processing in the brain network, is used to assess the diversity of brain regions participating in different functional communities. An increase in the entropy value implies that brain regions are involved in multiple functional communities, which not only reflects an enhancement of the diversity of network functions but also demonstrates the flexibility and complexity of brain activities. Conversely, a decrease in the entropy value indicates that the functions of brain regions are relatively homogeneous ([37]; [36]). Brain entropy can accurately capture these characteristics of brain activities, showing great potential in clinical applications. It provides a unique and valuable perspective for understanding the functional mechanisms of the brain, as well as for studying brain-related diseases and cognitive functions. Obsessive–compulsive disorder (OCD), as a complex mental disorder, has a very complicated neuropathological mechanism, which is highly likely to involve abnormal changes in the connection patterns among multiple brain regions. Traditional research methods often focus on the activities of individual brain regions (nodes) and their interconnections. However, this approach that only focuses on node activities has limitations and may not be able to comprehensively and deeply capture the abnormal characteristics of the brain networks in OCD patients. The emergence of the Edge Functional Connectivity (eFC) analysis method provides new ideas for studying the functions of brain networks. This method, from the unique perspective of connection edges, can explore the functions of brain networks in more detail. It serves as a powerful tool for revealing the abnormal mechanisms of the brain networks in OCD patients and helps us to have a deeper understanding of the neurobiological basis of OCD.

In this study, we applied edge-centric analysis to explore brain network interactions in a large prospective cohort of patients with interictal OCD. We hypothesized that OCD would be characterized by abnormal eFC or edge community overlap within brain networks. We used edge graph construction, *k*-means clustering analysis, the examination of component and network entropy, and community similarity analysis to comprehensively search for potential correlations with clinical symptoms.

## 2. Methods

### 2.1. Participants

This study recruited a total of 33 patients with obsessive–compulsive disorder (OCD) (24 males, 9 females) and 44 healthy control (HC) participants (24 males, 20 females), all of whom were of Han Chinese ethnicity. The participants were evaluated by experienced clinical psychiatrists and psychologists to ensure diagnostic accuracy and consistency.

OCD patients were recruited from Shenzhen Kangning Hospital and screened by psychiatrists using the Structured Clinical Interview for DSM-IV Axis I Disorders (SCID) to confirm that they met the diagnostic criteria for OCD as defined by the Diagnostic and Statistical Manual of Mental Disorders, Fourth Edition (DSM-IV). Patients meeting any of the following exclusion criteria were not included in the study: (a) age below 10 years or above 65 years; (b) history of traumatic brain injury, neurological disorders, or other major conditions that could affect neural function (e.g., epilepsy, stroke, or neurodegenerative diseases); (c) history of alcohol or substance abuse within 12 months prior to the study; (d) pregnant or breastfeeding women; (e) presence of severe psychiatric disorders, such as schizophrenia, bipolar disorder, or other conditions that significantly impair cognitive function.

All participants provided written informed consent after receiving a detailed explanation of the study. This study was approved by the Institutional Review Board (IRB) of Shenzhen Kangning Hospital.

### 2.2. Data Acquisition

We investigated edge-centric metrics in patients with OCD and HCs. We used data obtained during movie watching (an external stimulus task) to analyze brain activity features ([34]). The video clips are presented in a non-verbal form, conveying information through the actions, expressions of animated characters, and scene interactions. For example, they depict characters experiencing physical pain (such as being struck by lightning) or showing mental states (such as anxiety and surprise). The clips contain plots that evoke empathy (such as witnessing others’ pain) and theory of mind (such as inferring characters’ intentions), which can spontaneously induce neural activities related to the Pain Matrix and the Theory of Mind Network. The actions and scene changes in the animation (such as weather phenomena and character interactions) can activate the visual cortex, motor cortex, and limbic system, simulating neural responses in real-life situations.

Images were acquired using a 3.0-Tesla MR system (Philips Medical Systems Nederland B.V.) equipped with an eight-channel phased-array head coil. During scanning, participants were instructed to lie still with their eyes closed, remain awake, and avoid moving or focusing on specific thoughts. rs-fMRI data were collected using gradient-echo echo-planar imaging sequences [repetition time (TR) = 2000 ms, echo time (TE) = 30 ms, flip angle = 90°, field of view (FOV) = 220 × 220 mm, matrix = 64 × 64, number of slices = 33, slice thickness = 4.0 mm]. The scanning session for each participant lasted 480 s and yielded 240 whole-brain volumes. In addition, high-resolution T1-weighted anatomical images were acquired (TR = 8 ms, TE = 3.7 ms, flip angle = 7°, FOV = 240 × 240 mm, matrix = 256 × 256, slice thickness = 1.0 mm, voxel size = 1.0 × 1.0 × 1.0 mm^3^).

### 2.3. Functional Imaging Data Preprocessing

Data preprocessing was carried out using the Statistical Parametric Mapping toolbox (version 12; https://www.fil.ion.ucl.ac.uk/spm, accessed on 10 February 2025) and the Data Processing Assistant for rs-fMRI (version 4.4; http://rfmri.org/dpabi, accessed on 10 February 2025). The preprocessing steps were as follows: (1) data conversion from DICOM format; (2) discarding of the first 10 timepoints to eliminate signal instability; (3) correction for slice timing; (4) realignment of head motion and confirmation that no participant exceeded 1.5° head rotation or 1.5 mm translation; (5) registration of the functional images to the corresponding T1-weighted structural images; (6) out-regression of confounding factors, such as the six head motion parameters, white matter signals, and cerebrospinal fluid signals in the first-level analysis; (7) normalization of the functional data to the Montreal Neurological Institute stereotactic space and resampling to a voxel size of 3 × 3 × 3 mm^3^; (8) spatial smoothing of the images using a 6 mm Gaussian kernel; (9) application of a band-pass filter (0.01–0.08 Hz); and (10) performance of micro–head motion correction via the replacement of volumes with >0.5 mm frame-wise displacement. Global signal regression (GSR) was not performed, as studies have highlighted the importance of global signals in the examination of psychiatric disorders and the potential impact of GSR on anticorrelation, raising questions about its appropriateness ([64]; [53]).

### 2.4. Clinical Assessments

The Yale–Brown Obsessive–Compulsive Scale (Y-BOCS) ([27]) was used to assess the severity of patients’ obsessive–compulsive symptoms; separate obsession and compulsion scores were calculated. Using the revised Obsessive–Compulsive Inventory ([20]; [47]), obsessive–compulsive symptoms were classified as washing, obsession, hoarding, checking, neutralizing, and ordering. The three subscales of the Obsessive Beliefs Questionnaire-44 ([62]) [responsibility (obligation), perfectionism, and thought control] were used to assess patients’ obsessive beliefs. Depressive symptoms were assessed using the Beck Depression Inventory (BDI) ([33]), and state and trait anxiety symptoms were measured separately using the State–Trait Anxiety Inventory (STAI) ([51]).

### 2.5. Brain Parcellation and Intrinsic Connectivity Network Analysis

Two hundred independent components (ICs) were identified and categorized into 16 intrinsic connectivity networks ([38]): control networks A–C (ContA–C), default mode networks A–C (DefaultA–C), dorsal attention networks A and B (DorsAttnA, B), the limbic system, salience/ventral attention networks A and B, somatomotor networks A and B, the temporoparietal network, the visual central network, and the visual peripheral network. Spatial structural maps of these networks are provided as Figure 1, and network functions are listed in Table 1. We applied edge graph construction, *k*-means clustering, community overlap detection, and clinical correlation analysis to explore interactions between regions and examine the specificity of edge centrality results (Figure 2) ([18]).

#### 2.5.1. Edge Graph Construction

The time series of each IC obtained from group independent component analysis (GICA) was extracted, and an edge graph was constructed following the methods of [18] ([18]) and [13] ([13]). We calculated a *z* score for each time series and computed element-wise products for all series pairs (*z_i_* and *z_j_*). This calculation yielded vectors of length *T* whose elements encoded moment-by-moment co-fluctuation between ICs *i* and *j*, forming a co-fluctuation edge time series matrix. At any given timepoint, consistent increases or decreases in times series *i* and *j* relative to baseline yield positive co-fluctuation results, and changes in opposite directions yield negative results. For *N* time series, this results in $M=\frac{N(N-1)}{2}$ pairs of co-fluctuations, each with a length of *T*. The eFC between two pairs (i.e., *ij* and *uv*) was computed as follows:(1)$${eFC}_{ij,uv}=\frac{\sum_{t}c_{ij}(t)c_{uv}(t)}{\sqrt{\sum_{t}c_{ij}{\left(t\right)}^{2}}\sqrt{\sum_{t}c_{uv}{\left(t\right)}^{2}}}$$

Then, the M×M matrix was directly clustered using the *k*-means algorithm with standardized Euclidean distance. All edges associated with a specific edge (i.e., an entire row of the eFC matrix) were pooled to form single samples. We tested the number of clusters (k) ranging from 2 to 10 in increments of 1 and repeated the algorithm 250 times under random initial conditions.

#### 2.5.2. Entropy Calculation

Community entropy is a measure of edge overlap ([18]). When the distribution of edge community assignments is more uniform, the normalized entropy approaches 1; when an edge is assigned to a single community, this value approaches 0. Entropy was calculated by first evaluating the participation of brain region i in cluster c:(2)$$p_{ic}=\frac{1}{N-1}\sum_{j\neq i}\delta (g_{ij},c)$$
where for each edge {i,j}, i,j∈{1,…,N} is assigned to one of k clusters; $g_{ij}\in \{1,\ldots ,k\}$ represents the cluster assignment of the edge connecting ${IC}_{i}$ and ${IC}_{j}$; and $\delta (C_{ij},c)$ denotes the Kronecker delta value. By definition, $\sum_{c}p_{ic}=1$ and $p_{i}=[p_{i1},\ldots ,p_{ik}]$ are vectors representing probability distributions.

Thus, the entropy of the distribution reflects the extent to which the community affiliations of brain region i are distributed evenly across all communities or concentrated within a few. We calculated entropy using the following equation:(3)$$h_{i}=-\sum_{c}p_{ic}{log}_{2}p_{ic}$$

We calculated the normalized entropy of each IC obtained from GICA as a component-level value. Additionally, we computed the normalized entropy for each brain network to better understand systemic differences between the cohorts.

#### 2.5.3. Community Similarity Calculation

For the upper triangular eFC matrix of size N×N, $x_{ij}$ represents the edge community assignment between nodes i and j. For the element X in the *i*th column, $x_{i}=[x_{1i},\ldots ,x_{Ni}]$ represents the community labels of all edges in which node i participates. Therefore, we compared the edge community similarity between nodes i and j by calculating the similarity between vectors $x_{i}$ and $x_{j}$ (proportion of elements in the two vectors with the same community label):(4)$$s_{ij}=\frac{1}{N-2}\sum_{u\neq i,j}\delta (x_{iu},x_{ju})$$
where $\delta (x_{iu},x_{ju})$ is the Kronecker delta value ([18]). Increased community similarity between nodes i and j was taken to indicate that the respective functionally connected brain regions tended to be assigned to the same edge community, and vice versa.

### 2.6. Statistical Analysis

We investigated differences in eFC, entropy, and community similarity between cohorts using two-sample independent *t* tests and χ^2^ tests, conducted with SPSS 20.0 and MATLAB R2022b. Bonferroni correction for multiple comparisons was applied, with the significance threshold set at α = 0.05/n, where n represents the number of comparisons.

## 3. Results

### 3.1. Participants’ Demographic and Clinical Characteristics

The final analysis was conducted with data from 77 participants (44 patients with OCD and 33 HCs). No between-group difference in age (*p* = 0.789) or sex (*p* = 0.103) was identified (Table 2), eliminating the potential for confounding effects of these factors on the results. However, significant differences were observed between the OCD and HC groups in the number of years of education and Y-BOCS, BDI, and STAI scores (all *p* < 0.001), reflecting disparities in educational background, depression, anxiety, and the severity of obsessive–compulsive symptoms between patients with OCD and HCs, further highlighting the multidimensional clinical distinctions of OCD. These variables were thus treated as confounders and were controlled for in subsequent analyses. 

### 3.2. Group Differences in eFC and Entropy

Results corrected for multiple comparisons revealed no significant difference in eFC between the OCD and HC groups. However, significant differences in entropy were observed (Figure 3). Entropy values for the ContA, ContC, and DefaultB networks were significantly higher in the OCD group than in the HC group (*p* = 0.0016, 0.0017, and 0.0002, respectively). As higher entropy values are generally associated with more complex brain activity and diversity of information processing, these results suggest that patients with OCD have more complex neural activity in these networks. Conversely, entropy values for the DorsAttnA network were higher in the HC group than in the OCD group (*p* = 0.0302). This result may reflect more information processing and/or a greater cognitive load in this network in healthy individuals. This difference suggests that functional regulation patterns in the DorsAttnA network are distinct between patients with OCD and HCs.

Overall, although eFC did not differ between the OCD and HC groups, the observed differences in entropy suggest that patients with OCD exhibit distinct information processing characteristics in specific brain networks. Further research is warranted to explore the implications of these differences in the neurobiological basis of OCD and whether they are associated with symptom presentation or underlying pathological mechanisms. Spatial maps of the ContA, ContC, DefaultB, and DorsAttnA networks are provided as Figure 4.

## 4. Discussion

In this prospective study, edge centrality metrics were used to examine functional abnormalities in the brain networks of patients with OCD. These patients exhibited significant FC abnormalities in the control network, DMN, and dorsal attention network, with greater entropy in the former two and lesser entropy in the latter relative to HCs, potentially reflecting more disordered brain activity (information processing) patterns in these networks. Given that we controlled for the potential confounding effects of certain covariates, these results are reliable and accurate. These findings provide new insights into the neurobiological mechanisms of OCD and may offer theoretical support for future clinical intervention strategies.

Structural and functional abnormalities in the thalamus and brain networks, such as the ContA, ContC, DefaultB, and DorsAttnA networks, in patients with OCD have been explored in many node centrality and biomarker studies. In a study in which rs-fMRI was used to analyze the patterns of endogenous-blood-oxygen-level-dependent signal fluctuations, patients with OCD showed FC abnormalities in the control network, including reduced connectivity in the posterior temporal region, increased connectivity in other control regions, and significantly more local clustering, relative to HCs ([67]). Another rs-fMRI study revealed FC abnormalities between the frontoparietal network and DMN in patients with OCD, potentially related to symptom phenotypes, such as difficulties in disengaging from internal thoughts and focusing on external tasks ([67]). Another study revealed abnormal event-related potential components in patients with OCD performing specific tasks, suggesting that an over-concentration of attentional resources forms the neurophysiological basis of the core symptoms of OCD ([57]).

The control network is involved primarily in executive functions, cognitive control, behavioral inhibition, and self-monitoring abilities and is related closely to cognitive processes, such as situational adaptation, planning, and decision making. The control network (especially ContA and ContC) is primarily involved in higher-order cognitive control, behavioral inhibition, and situational adaptation. Elevated entropy values indicate increased complexity of information processing in these networks, potentially reflecting disorganization in cognitive regulation among OCD patients. For example, ContA is associated with decision making and conflict monitoring, and its increased entropy may correspond to patients’ difficulty in inhibiting compulsive behaviors. Meanwhile, ContC is involved in executing complex cognitive tasks, and its abnormality may relate to perseveration in repetitive behaviors. In contrast, ContB, which is more engaged in basic cognitive tasks (e.g., working memory), did not show significant alterations in OCD, suggesting differential roles of distinct control subnetworks in pathological mechanisms. This finding supports the “cognitive control dysfunction model” of OCD, where dysregulation of prefrontal–striatal circuits leads to patients’ inability to flexibly adjust behavioral patterns ([16]; [59]). The control network functional patterns in these patients differ significantly from those in HCs, which may form the neurophysiological basis for the emotional and behavioral regulation disorders typifying and provide potential targets for clinical intervention ([25]).

The default mode network (DMN) is typically active during rest and is involved in self-related thinking and introspection. The DefaultB network is responsible for self-referential thinking and monitoring of internal mental states. Elevated entropy values in this network may indicate more fragmented information integration in OCD patients, which is associated with excessive self-reflection and intrusive thoughts. This aligns with the “DMN hyperactivity hypothesis” ([46]), suggesting that abnormal DefaultB activity may trap patients in compulsive introspection, making it difficult to shift attention from internal thoughts to external tasks. Additionally, abnormal functional connectivity between DefaultB and the posterior cingulate gyrus may further exacerbate the perception of compulsive symptoms ([7]). This perspective is also supported by the systematic review and meta-analysis conducted by [58] ([58]), who reported significant abnormalities in activation patterns of relevant brain regions during inhibitory control tasks in OCD patients, potentially linked to dysfunction in the DefaultB network.

The dorsal attention network plays crucial roles in spatial attention and goal-directed behavior regulation, helping individuals respond to environmental changes and flexibly shift their attention focus. DorsAttnA is responsible for orienting and maintaining spatial attention. Reduced entropy suggests inflexible information processing patterns in this network, which may correspond to OCD patients’ excessive focus on specific stimuli (e.g., compulsive checking or cleaning behaviors) and deficits in attentional switching ([44]). This “attentional fixation” phenomenon could arise from desynchronization between the dorsal attention network and prefrontal regulatory circuits, leading to impaired cognitive resource allocation and exacerbating symptom persistence ([2]). As a result, these patients may struggle to adapt to environmental changes, which significantly impacts their emotional regulation and cognitive functions. This dysfunction in the dorsal attention network may be a key factor in the persistence and exacerbation of obsessive–compulsive symptoms. Its understanding may provide valuable insights for future clinical interventions, helping to improve the attention control and cognitive flexibility of patients with OCD.

### Limitations

This study has several limitations: First, we did not explore potential differences in brain network functions among patients with different OCD subtypes. Future studies should involve the exploration of differences in brain network structure and function among individuals with different subtypes of OCD. Additionally, the study sample was relatively small, which may have affected the generalizability of the results. To validate the findings of this study, larger samples should be recruited for future research. Finally, although we controlled for some potentially confounding variables, uncontrolled factors (e.g., medication use and emotional states) may have influenced the results. Thus, future studies should involve stricter control of these potential confounders.

## Figures and Tables

**Figure 1 behavsci-15-00488-f001:**
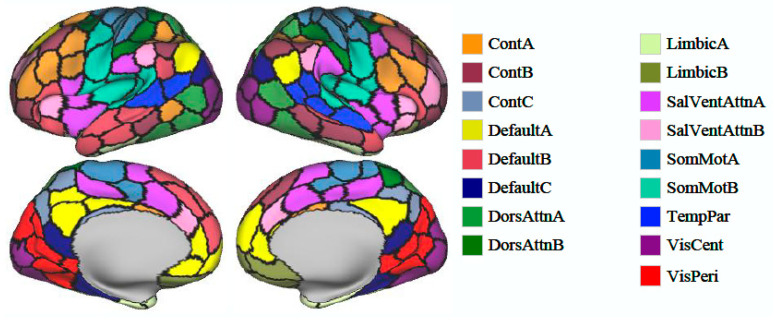
Spatial maps of 200 identified independent components divided into 16 intrinsic brain networks.

**Figure 2 behavsci-15-00488-f002:**
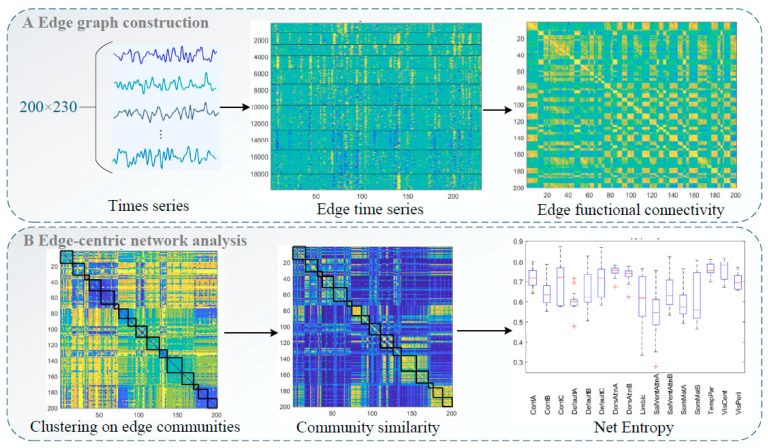
Flow of edge-centric functional network analysis: (**A**) Edge graph construction and edge functional connectivity calculation. (**B**) Edge-centric network analysis, including edge community clustering and entropy and community similarity calculations.

**Figure 3 behavsci-15-00488-f003:**
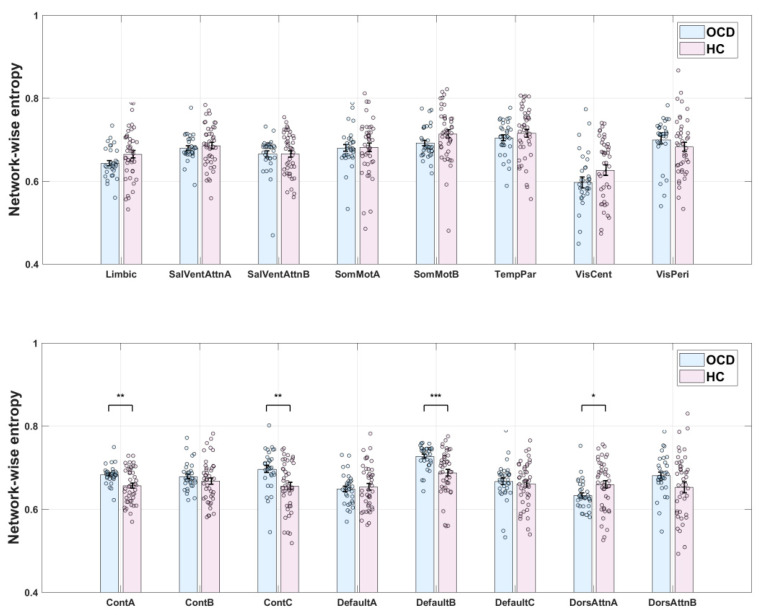
Network entropy in patients with obsessive–compulsive disorder (OCD) and healthy control s (HCs). *: Indicates *p* < 0.05; **: Indicates *p* < 0.01; ***: Indicates *p* < 0.001.

**Figure 4 behavsci-15-00488-f004:**
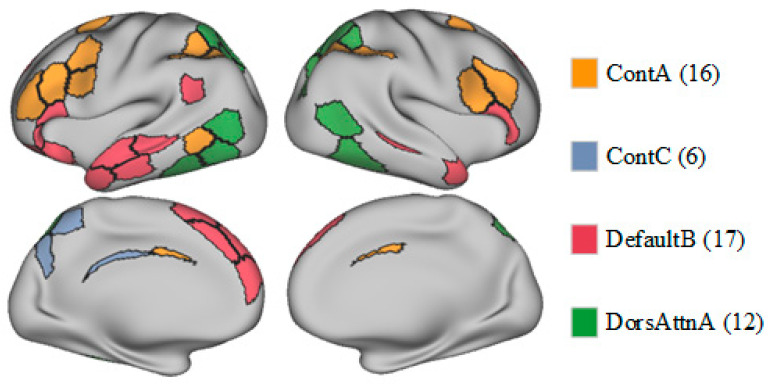
Spatial maps of the ContA, ContC, DefaultB, and DorsAttnA networks.

**Table 1 behavsci-15-00488-t001:** Brain networks and their primary functions.

Network	Function
Visual central network	Primarily the processing of central visual information
Visual peripheral network	Processing of peripheral visual information
Somatomotor network A	Involved in bodily sensation and motor control
Somatomotor network B	Related to bodily sensation and movement, potentially involving more specific motor or sensory regions
Dorsal Attention network A	Involved in the allocation and control of spatial attention
Dorsal Attention network B	Similar to network A, responsible for various attention tasks
Salience/ventral attention network A	Processng of the salience of external stimuli
Salience/ventral attention network B	May be involved in emotional and stimulus response processing
Limbic network	Associated with emotional processing and memory functions, handling of emotions and autonomic functions
Control network A	Cognitive control, decision making, and executive functions
Control network B	Similar to control network A, involved in various cognitive control tasks
Control network C	Associated with complex cognitive tasks and executive functions
Default mode network A	Involved in introspection, memory recall, and social cognition
Default mode network B	Involved in self-referential cognitive processes
Default mode network C	Involved in broader internal thought processes and memory
Temporoparietal network	Involved in social cognition, language processing, and multisensory integration

**Table 2 behavsci-15-00488-t002:** Demographic and clinical characteristics of patients with obsessive–compulsive disorder (OCD) and healthy controls (HCs).

Characteristic	OCD	HC	*p*
*n*	33	44	-
Age (years)	21.77 ± 7.79	21.11 ± 5.09	0.789
Sex (female)	9	20	0.103
Onset age	16.52 ± 5.66	NA	-
Education (years)	11.17 ± 1.41	13.95 ± 4.24	<0.001
Y-BOCS score	12.26 ± 6.54	11.25 ± 5.35	<0.001
BDI score	11.47 ± 11.41	10.99 ± 4.89	<0.001
STAI score	86.43 ± 23.86	84.50 ± 14.44	<0.001

Values are means ± standard deviations.

## Data Availability

The datasets generated and analyzed during the current study are not publicly available due to privacy concerns and ethical restrictions, but are available from the corresponding author upon reasonable request.

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
