# Peer review of "Brain Network Abnormalities in Obsessive–Compulsive Disorder: Insights from Edge Functional Connectivity Analysis"

_behavsci, 2025, doi:10.3390/bs15040488_

Round 1

Reviewer 1 Report

Comments and Suggestions for Authors

The aim of this manuscript is to explore the abnormalities of brain networks in patients with obsessive-compulsive disorder (OCD). It uses the edge functional connectivity (eFC) analysis method to deeply analyze the differences between OCD patients and healthy controls, with the goal of uncovering the potential neurobiological mechanisms of OCD. This is an area of interest to the journal. The study collected functional and structural brain imaging data from 44 OCD patients and 33 healthy controls while they were watching a movie. Through a series of rigorous data preprocessing steps, clinical assessments, brain parcellation, and complex eFC analysis processes, including edge graph construction, entropy calculation, community similarity calculation, and statistical analysis. The results show that, after controlling for potential confounding variables, there are significant differences in the entropy values of some brain networks between OCD patients and healthy controls. Specifically, the entropy values of the control networks (ContA, ContC) and the default mode network (DefaultB) in OCD patients are significantly higher than those in healthy controls, while the entropy value of the dorsal attention network (DorsAttnA) is significantly lower. However, there are no significant differences in eFC between the two groups. Based on these findings, the study speculates that the changes in the entropy values of these brain networks may be closely related to the core symptoms of OCD patients, such as attention deficits, impaired cognitive control, and self-referential thinking. The manuscript is well-written, and the analysis addresses the issues of interest. However, there are some issues that need to be clarified in the manuscript.

  1. When exploring the relationship between brain network abnormalities and OCD, the title of the paper mentions "Edge Functional Connectivity Analysis". Why was the eFC method used? What are the advantages of this method compared to previous ones? In the main text, only a simple comparison of its differences from traditional node connectivity methods is made, and it does not elaborate on the unique contributions of this method in studying the brain networks of OCD patients compared to other cutting-edge methods.
  2. The selection of the k-value range (2 to 10) and the number of repetitions (250) in the k - means clustering algorithm in eFC analysis lacks justification. Why was this specific range selected, and how was it determined to yield optimal clustering? For example, could higher k-values (e.g., 15) better capture brain network communities? In addition, without comparison with other clustering algorithms (e.g., hierarchical clustering), how can the authors ensures the superiority of the k - means algorithm in this research?
  3. In the data collection part of the study, silent clips from "Partly Cloudy" were used as stimulus materials to obtain brain activity data. Could you briefly describe the data content of this movie clip? And compared with traditional data collection methods (such as resting-state data collection), how does this information affect the brain's response pattern?
  4. When exploring the relationship between the control network, the default mode network, and the dorsal attention network and OCD, the analysis of the functional differences of these networks in normal people and OCD patients is not comprehensive enough. The conclusion is drawn only based on the differences in entropy values, without fully integrating the functional characteristics of these networks in multiple aspects such as cognition and emotion in existing research to deeply analyze their internal connections with the core symptoms of OCD, resulting in a slightly insufficient explanatory power of the conclusion.
Comments on the Quality of English Language

I feel that the quality of english language for this manuscript is good.

Reviewer 2 Report

Comments and Suggestions for Authors

The manuscript "Abnormal Brain Network Characteristics in OCD: Insights from Edge Functional Connectivity" explores novel edge-centric metrics to investigate functional connectivity abnormalities in key brain networks (DAN, CN, DMN) in OCD patients. The use of a movie-watching paradigm is an interesting methodological choice, and the findings contribute to our understanding of OCD-related neural mechanisms. However, some methodological and interpretational aspects require clarification.

Major Concerns:

  1. While the study introduces eFC as an alternative to traditional node-based connectivity approaches, the conceptual and physiological significance of network entropy remains unclear. Specifically, what does increased or decreased entropy in ContA, ContC, DefaultB, and DorsAttnA indicate about cognitive processes in OCD? More theoretical grounding on how entropy changes reflect network function would improve interpretation.
  2. The Discussion does not sufficiently explain why entropy changes are specific to ContA, ContC, DefaultB, and DorsAttnA networks. For example, why is ContC affected while ContA is not? Further discussion linking these findings to established OCD models (e.g., cognitive control dysfunction, DMN overactivity) would strengthen theoretical relevance.
  3. In the Discussion, the manuscript references conflicting literature (e.g., DAN/FPN hyperconnectivity vs. hypoconnectivity) but does not reconcile these discrepancies with the current findings.

Minor Points:

  1. The terminology used to describe brain networks is inconsistent throughout the manuscript. For example, the Control Network is referred to as "CN" in some sections and as separate components (ContA, ContB, ContC) in others. Standardizing these terms would improve clarity.
  2. Figure 3 (network-wise entropy differences) lacks an explanation of the error bars.
  3. Some citations are outdated (e.g., Beck et al., 1961 for BDI; Speilberger et al., 1983 for STAI).
Comments on the Quality of English Language

The language and grammar need some work.

Reviewer 3 Report

Comments and Suggestions for Authors

The authors adopted the edge functional connectivity (eFC) analysis method to analyze the interactions of brain networks, providing a new perspective for researching the neurobiological mechanisms of obsessive - compulsive disorder (OCD). The study not only calculated network entropy but also conducted community similarity analysis, exploring the complexity and dynamics of the brain networks in OCD patients from multiple perspectives. By involving multiple intrinsic connectivity networks (ICNs) and describing their functions in detail, the comprehensiveness of the research was enhanced. Well - recognized clinical assessment tools such as the Yale - Brown Obsessive - Compulsive Scale (Y - BOCS), Beck Depression Inventory (BDI), and State - Trait Anxiety Inventory (STAI) were used to ensure a detailed record of the clinical characteristics of the participants. Potential confounding factors such as depression and anxiety scores were controlled in the analysis, enhancing the robustness of the results and reducing the likelihood of false positives. The topic of this study is interesting, with a relatively large sample size and reasonable analysis methods. Here are my insights:

  1. Please explain why an increase in entropy in certain networks may be associated with more disordered brain activity in OCD patients?Also, explain how this is related to clinical symptoms? The paper mentions that "higher entropy = more complex information processing", but it is necessary to supplement the literature basis to support this hypothesis.
  2. In the article, 200 independent components are divided into 16 intrinsic connectivity networks, mainly based on existing research. However, hasthis division method in the samples of this study been tested? Directly using the existing division method may not accurately reflect the brain network situations of OCD patients and healthy controls in this study. In addition, individual differences in brain network division are not considered—might this oversimplification bias functional abnormality analyses?
  3. In the article, some potential confounding variables such as age, gender, educational level, depression, and anxiety scores were controlled. Are there any other factors that may affect the results? For example, medication use, life stress events, sleep quality, etc.
  4. In the results section, the study reported significant results. However, in the conclusion section, is there more sufficient explanation to prove the potential mechanisms between network abnormalities and specific OCD symptoms?

Reviewer 4 Report

Comments and Suggestions for Authors

Dear Editors,

I have thoroughly reviewed the manuscript. Even if not entirely new, I really liked the joint eFC and entropy approach. The methodological idea is interesting, and entropy has been previously studied as a metric to evaluate neural correlates in dysfunctional brain networks in Alzheimer’s and schizophrenia, but it is quite novel in OCD research. The manuscript is well-written and clearly presented in most of its parts. I also appreciated the limitations section of the discussion. However, I have some concerns that I would like to see addressed, and I think some work is still needed before the manuscript is ready for publication. 

Detailed comments

State of the art and References

The references are inadequate and should be substantially improved. A significant amount of literature regarding FC and OCD is surprisingly absent. Similarly, there are no references to studies that looked at how behavior, functional connectivity, entropy, and complexity measures are affected in humans and animals during tasks testing the cognitive abilities impaired during OCS. Such abilities include cognitive flexibility, response inhibition, and visuospatial memory; appropriate tasks to measure them are, for example, the stop-signal task, the go/no-go task, and the delayed non-matching to sample task. These tasks provide us objective information about how OCD affects cognitive functions and are critical for elucidating the behavioral and neural correlates of cognitive neuropsychological processes. Since it's known that DAN dysfunction affects attentional shifting, that CN is involved in executive function and response inhibition, and that DMN hyperconnectivity is present in OCD, they need to be included and partly discussed to contextualize the CN entropy findings in a wider framework and with existing works. Indeed, the author’s findings could be relevant to research on attention control and response inhibition. Here are some of them. 

Harrison et al, 10.1001/archgenpsychiatry.2009.152

Gonçalves et al, https://doi.org/10.1038/srep44468

Uhre et al, https://doi.org/10.1016/j.nicl.2022.103268

Wheaton et al, https://doi.org/10.1016/j.jbtep.2025.102019 

Buschman and Miller, https://doi.org/10.1126/science.1138071

Posner et al, https://doi.org/10.1002/hbm.23408

Liu et al,  https://doi.org/10.1002/hbm.26457 

M Prabhavi N Perera et al, https://doi.org/10.1093/cercor/bhae327 

McLaughlin et al, https://doi.org/10.1017/S1355617716000540

Bardella et al, https://doi.org/10.1162/netn_a_00365

Aron, https://doi.org/10.1523/JNEUROSCI.3644-07.2007

Diesburg and Wessel, https://doi.org/10.1016/j.neubiorev.2021.07.019

Jana et al, 10.3390/brainsci11050607

Criaud et al, https://doi.org/10.3389/fpsyg.2012.00059

Varley et al, https://doi.org/10.1073/pnas.2207677120

Zhai et al, https://doi.org/10.1093/nsr/nwad312

Bosc et al, https://doi.org/10.1038/srep45267

Tomiyama et al, https://doi.org/10.1002/hbm.25699

Jana and Aron, 10.1177/09567976211055371

Maatoug et al,  https://www.nature.com/articles/s41398-019-0667-3

Grant et al, https://doi.org/10.1523/jneurosci.0816-21.2022

Maatoug et al, https://doi.org/10.1038/s41398-020-0735-8

Ghani et al, https://doi.org/10.1016/j.neubiorev.2020.07.020

Pickenhan e Milton, https://doi.org/10.3758/s13415-023-01153-w

Ruchsow et al., https://doi.org/10.1007/s00702-007-0779-4

M Prabhavi N Perera et al (2023), http://dx.doi.org/10.1017/S0033291723000843 

Yu et al, https://doi.org/10.1016/j.bpsc.2024.12.001

Koprivova et al, https://doi.org/10.1016/j.clinph.2011.01.051

Masharipov et al, https://doi.org/10.3390/ijerph20021171 

Brown et al, https://doi.org/10.3389/fnsys.2015.00124

Ravindran et al, 10.1017/S0033291719001090 

Kepecs et al, https://doi.org/10.1098/rstb.2012.0037

Siddeswara et al, https://doi.org/10.1016/j.ajp.2021.102857

Cui et al, https://doi.org/10.3389/fpsyt.2020.00098

Methodological Limitations

As also stated by the authors, the study presents some methodological limitations that constrain the interpretation of its findings. Aside from the small sample size, the study doesn't differentiate between different types of symptoms (like checking, contamination, or hoarding), which may have different effects on the brain. Additionally, the use of a movie-watching paradigm could introduce variability in cognitive and emotional engagement, potentially confounding functional connectivity measures. Furthermore, it would be interesting to see how some of the tasks above would affect the authors' metrics. I’m not saying the authors should perform new experiments or analysis, but it would be interesting for them to speculate on what effects they would expect in their metrics while performing some of the tasks mentioned above  Future studies should incorporate larger, stratified cohorts and controlled task paradigms to ensure robust conclusions. This point should be stressed more. 

Currently, the authors are somehow overstating the interpretational reach of their findings, particularly regarding the role of network entropy as a marker of OCD pathology. The results are correlational and do not establish whether entropy differences are causally linked to OCD symptoms. Furthermore, similar entropy alterations have been observed in other psychiatric conditions, suggesting that the findings may reflect a general marker of dysregulated cognitive control rather than an OCD-specific signature. Without longitudinal validation or predictive modeling, it is premature to assert that network entropy holds diagnostic or prognostic significance for OCD. Hence, It is highly premature to propose network entropy as a biomarker for OCD, as the study lacks longitudinal validation, classification accuracy analyses, and treatment-response correlations. I would like the authors to mediate on this point. 

The Introduction section, while providing general information on OCD prevalence and impact, is unfocused and lacks direct relevance to the study’s objectives. Instead, it should offer a precise rationale for using entropy and eFC methods and cognitive abilities affected by OCD. Similarly, the Discussion section should critically reflect on the limitations of entropy as a clinical tool, rather than implying its immediate translational relevance. A more focused manuscript structure would improve clarity, rigor, and impact.

Round 2

Reviewer 4 Report

Comments and Suggestions for Authors

Review Report for The Manuscript  behavsci-3503493_2nd_Round

Dear Editors,

I appreciate the authors' efforts to improve the manuscript and incorporate the suggested revisions. The authors' response regarding the remodulation of some overstatements and interpretations was also appreciated. The authors also improved the presentation by making it more focused. However, the manuscript still contains some inaccuracies, and some points need to be addressed before it is ready for publication. Addressing such concerns will enhance the completeness, accuracy and clarity of the manuscript. This should not require significant time, and I am confident the author can reasonably take this latest step to enhance the quality of their work.

Detailed comments

The references require further refinement. Although the authors have included some of the suggested references, others remain absent and should be added to provide a broader and appropriate context. A work on such a broad topic as FC and entropy metrics related to a disorder such as OCD, which includes deficits in some key cognitive domains such as motor initiation, execution and inhibition; spatial working memory; and spatial recognition, cannot contain only 50 Refs (before they were even less, 38). Key ones to add are listed below, organized by the authors' PtP response themes. Especially the part about some crucial cognitive abilities affected by OCD, such as response inhibition, proactive and reactive control, etc.., needs to be improved. When necessary, (e.g, for response inhibition, working memory and related network studies), we encourage authors to expand these lists with related work from the specific groups. Especially an important paradigm such as the Stop-signal task is mentioned very briefly, e.g., in the sentence “functional connection strength[..] on the efficiency of response inhibition in the stop-signal task”  should be detailed more. Related to this, the following description of reference 44 is inaccurate and requires correction: “dynamic causal model study by Varley et al. [44]” . Dynamic casual model is an imprecise and vague term here, and it does not convey the message and the importance of the study properly. In addition, the description does not include the reference to another important and very similar study  that was recommended along with Ref 44 in the previous round (that is: Bardella et al, https://doi.org/10.1162/netn_a_00365). I recommend revising it with something along these lines: “Both studies quantified neural information processing dynamics via entropic measures (e.g.,transfer entropy ) to characterize node and edge-level network changes and adaptations in response to cognitive demands. These findings are highly relevant for disorders affecting cognitive control, such as OCD, ADHD [etc.. ], where deficits in response inhibition and network adaptability contribute to impaired executive function, compulsivity, and motor dysfunction, highlighting potential neural correlates for cognitive impairment. “
Here’s some of the overlooked references mentioned above.

Behavior and cognitive abilities, in human and animal models

  • McLaughlin et al, https://doi.org/10.1017/S1355617716000540 
  • Masharipov et al, https://doi.org/10.3390/ijerph20021171 
  • Bosc et al, https://doi.org/10.1038/srep45267 
  • Marc et al., https://doi.org/10.3389/fnhum.2023.1106298  
  • Giuffrida et al, https://doi.org/10.3389/fpsyg.2023.1125066  
  • Pani et al, https://doi.org/10.1016/j.ridd.2013.06.032 
  • Aron, https://doi.org/10.1523/JNEUROSCI.3644-07.2007 
  • Criaud et al, https://doi.org/10.3389/fpsyg.2012.00059 
  • Zhai et al, https://doi.org/10.1093/nsr/nwad312 
  • Tomiyama et al, https://doi.org/10.1002/hbm.25699 
  • Ghani et al, https://doi.org/10.1016/j.neubiorev.2020.07.020
  • Maatoug et al,  https://www.nature.com/articles/s41398-019-0667-3 
  • Pickenhan e Milton, https://doi.org/10.3758/s13415-023-01153-w 
  • Ruchsow et al., https://doi.org/10.1007/s00702-007-0779-4
  • Brown et al, https://doi.org/10.3389/fnsys.2015.00124

Network analysis (in related cognitive abilities in health and/or dysfunction):

  • Yu et al, https://doi.org/10.1016/j.bpsc.2024.12.001 
  • Koprivova et al, https://doi.org/10.1016/j.clinph.2011.01.051
  • Grant et al, https://doi.org/10.1523/jneurosci.0816-21.2022 

FC  and OCD

  • Liu et al,  https://doi.org/10.1002/hbm.26457 
  • Stern et al., https://doi.org/10.1016/j.pscychresns.2016.08.006
  • Cui et al, https://doi.org/10.3389/fpsyt.2020.00098
  • Ravindran et al, 10.1017/S0033291719001090 
